# Information given by websites selling home self-sampling COVID-19 tests: an analysis of accuracy and completeness

Sian Taylor-Phillips ,[1,2] Sarah Berhane,[2,3] Alice J Sitch,[2,3] Karoline Freeman ,[1,2] Malcolm James Price,[2,3] Clare Davenport,[2,3] Julia Geppert,[1] Isobel M Harris ,[2] Osemeke Osokogu,[1] Magdalena Skrybant,[2,4] Jonathan J Deeks[2,3]

[1]Warwick Medical School, University of Warwick, Coventry, UK
[2]Institute of Applied Health Research, University of Birmingham, Birmingham, UK
[3]NIHR Birmingham Biomedical Research Centre, University Hospitals Birmingham NHS Foundation Trust and University of Birmingham, Birmingham, UK
[4]NIHR Applied Research Collaboration West Midlands, University of Warwick, Coventry, UK

**Correspondence to**
Dr Sian Taylor-Phillips;
s.taylor-phillips@warwick.ac.uk

## ABSTRACT

**Objectives** To assess the accuracy and completeness of information provided by websites selling home self-sampling and testing kits for COVID-19.

**Design** Cross-sectional observational study.

**Setting** All websites (n=27) selling direct to user home self-sampling and testing kits for COVID-19 (41 tests) in the UK (39 tests) and USA (two tests) identified by a website search on 23 May 2020.

**Main outcome measures** Thirteen predefined basic information items to communicate to a user, including who should be tested, when and how testing should be done, test accuracy, and interpretation of results.

**Results** Many websites did not provide the name or manufacturer of the test (32/41; 78%), when to use the test (10/41; 24%), test accuracy (12/41; 29%), and how to interpret results (21/41; 51%). Sensitivity and specificity were the most commonly reported test accuracy measures (either reported for 27/41 [66%] tests): we could only link these figures to manufacturers' documents or publications for four (10%) tests. Predictive values, most relevant to users, were rarely reported (five [12%] tests reported positive predictive values). For molecular virus tests, 9/23 (39%) websites explained that test positives should self-isolate, and 8/23 (35%) explained that test negatives may still have the disease. For antibody tests, 12/18 (67%) websites explained that testing positive does not necessarily infer immunity from future infection. Seven (39%) websites selling antibody tests claimed the test had a CE mark, when they were for a different intended use (venous blood rather than finger-prick samples).

**Conclusions** At the point of online purchase of home self-sampling COVID-19 tests, users in the UK are provided with incomplete, and, in some cases, misleading information on test accuracy, intended use, and test interpretation. Best practice guidance for communication about tests to the public should be developed and enforced for online sales of COVID-19 tests.

## Strengths and limitations of this study

► We believe this is the first research on accuracy of information provided by websites selling tests for COVID-19, where users may put themselves or others at increased risk of transmission if results are misinterpreted.

► We duplicated processes of searching and data extraction to minimise bias.

► Using pre-specified criteria, we found evidence that websites selling home self-sampling COVID-19 tests provided incomplete and inaccurate information on test accuracy and interpretation of test results at the point of purchase.

► We developed basic guidance on what should be communicated when selling tests, including the type of test; situations when the test should be used; the time when the test should be done and details of how it should be done; the name of the test and details from clinical accuracy studies; evidence of compliance with regulatory approvals; explanation of test results using accessible and relevant metrics such as predictive values; and guidance to the interpretation and actions based on results.

► We only included websites from the UK and USA, so while the principles of what should be communicated apply to all countries, the results about data completeness are not generalisable beyond the UK and USA.

## INTRODUCTION

The 2019 novel coronavirus (COVID-19) pandemic has resulted in national population measures such as restricted movement ('lockdown'), and mass testing programmes. Testing is regarded as critical to manage the pandemic – the two main test types available being molecular virus tests (to detect current infection) and antibody tests (to detect previous infection). The WHO recommends polymerase chain reaction (PCR)-based molecular virus testing of symptomatic individuals to detect current COVID-19 infection,[1] to enable identification and isolation of confirmed cases, and tracing of those exposed for further testing. However, due to the sensitivity for a single PCR test being as low as 70%,[2] the WHO states that even two consecutive negative PCR tests do not rule out infection with COVID-19.[1] Antibody tests are not recommended for individual use by

the WHO, because we do not yet understand whether the presence of antibodies infers immunity from future infection. Their sensitivity has been estimated at around 80%–90%, thus there is also a risk of false negatives.[3] Timing of testing is critical for both tests: molecular tests are thought to be most accurate when used within 5 days of the onset of symptoms,[4] antibody tests are most accurate 2 or more weeks after onset of symptoms.[3]

There are now multiple websites selling both molecular virus tests and antibody tests outside of national testing programmes. To ensure appropriate use, interpretation and actions following testing, it is necessary for tests to be sold with clear communication about who should use each test, when and how samples should be taken, and the implications of positive and negative results. Previous research investigating direct to user sales of genetic testing found that the information provided was incomplete, particularly the implications of test results and limitations of testing, and was not always in an accessible and understandable format.[5–7]

Direct to user sale of tests are regulated by the Medicines and Healthcare products Regulatory Agency (MHRA) in the UK and the Food and Drug Administration (FDA) in the USA. Europe[8] and the USA[9] operate a risk-based regulation for in vitro diagnostic devices (IVDs) which depends on the intended use of the test and indications for use. IVDs for home testing fall into higher risk categories reflecting the fact they are initiated, performed, and interpreted without professional guidance and require evidence that lay users correctly use the test and understand the test results. Lay user studies are required as the basis for the instructions for use (IFU) document for the IVD.[10] *Home sampling* tests are different from *home testing* as they receive approval based on home collected specimens with the test analysis being undertaken by professionals. At the time of writing, there were no COVID-19 antibody tests with a CE mark for either home sampling or home testing[11] (the two COVID-19 antibody tests purchased by the UK Government are approved for use in venous but not finger-prick blood samples) while several molecular virus tests have regulatory approval for home sampling and are being used in the UK track and trace programme.[12] Most websites selling COVID-19 tests would be classified by the MHRA as 'distributors', which gives clear obligations to supply the information provided by manufacturers with the test, but no specific guidance around communication on the website at the point of sale. Such claims are covered by the Advertising Standards Agency. In the US, there are no COVID-19 antibody tests with regulatory approval for home testing but four molecular (PCR) virus tests that have approval for home *sampling*[13] where the appropriateness of the test purchase is assessed by a professional either pre-purchase or following a purchase request.

We analysed the information given to individuals considering purchasing a molecular virus or antibody COVID-19 test online for home self-sampling. We chose to review tests for sale in both the UK and USA to cover two different regulatory systems with contrasting health services. We recorded information regarding who should be tested and when, claims about test accuracy, and information about how to interpret results. As the MHRA instigated a withdrawal of sales of antibody tests based on finger-prick blood samples on 29 May 2020 where tests require venous blood samples,[11] we also evaluated how test vendors have responded.

## METHODS
Our research question was how complete, accurate, and informative is the information that online websites selling home self-sampling and testing kits for COVID-19 provide to the public?

### Identification of websites
The search was designed to identify a representative sample of websites and online advertisements which would be seen by an individual searching for a non-specific COVID-19 test. We aimed to identify websites selling home self-sampling and testing for COVID-19 using molecular virus and/or antibody tests directly to users. Two researchers performed the searches independently on the same day (23 May 2020) using the Google search engine in incognito mode in Google Chrome, with geo-locations for the UK and for the USA. In order to emulate a simple search for a non-specific coronavirus test, the search terms were (coronavirus OR covid-19 OR covid19) AND (test OR testing OR kit). Two researchers independently screened all results against the inclusion criteria, and disagreements were resolved by a third researcher. For the UK search, we included websites moved to the top of the search results through advertisements, in order to mirror what a user would have seen on that day.

### Inclusion criteria
We included websites selling molecular virus and/or antibody tests for COVID-19 direct to users in either the USA or UK. We included point-of-care and laboratory-based tests, with the proviso that the sample was taken at home by the individual themselves. We excluded tests with assisted sampling (eg, drive-through testing), or where part of the testing process before purchase included video, telephone, or in-person contact with a medical professional (as we could not objectively assess the information content of such interactions). We included websites selling tests both via direct purchase and insurance funding, but excluded local or national government websites providing tests (including Public Health England [PHE] and the UK National Health Service [NHS]), and websites providing tests as part of a research study. We included all eligible tests, including where a single website sold multiple eligible tests. We excluded websites with a minimum order of more than a single test, as these targeted suppliers rather than individual users.

## Data extraction

We extracted information about the test manufacturer and type of test; when testing was recommended; claims made about test accuracy; the advice given about changing behaviour in light of test results; accreditation; and the test cost. We assessed the information provided against a predefined list of items which we would expect to be communicated to a person considering purchasing a test for COVID-19, detailed in table 1.

We extracted claims made about regulatory approval of the tests, in particular CE-IVD approval in the UK and FDA approval in the USA, and where possible compared claims to the actual approval status for the test. We also extracted claims made about approval from non-regulatory bodies such as PHE and the NHS.

Website contents were extracted between 23 and 28 May 2020. One researcher extracted data from each website onto a predefined data extraction form, and downloaded the website as a pdf file. A second researcher checked each extraction using the pdf copy to exclude temporal changes.

## Patient and public involvement

A public contributor (MS), with both experience of being involved in research and leading public involvement in research, provided input into this project. MS has an interest in communicating scientific information to lay audiences. The rapid timeframes in which the research was conducted limited the scope for more comprehensive public involvement. MS contributed to discussions and paper drafts and is included as a co-author.

Ethics approval was not required for this review of publicly available documents.

## RESULTS

For the UK our Google searches retrieved 550 results, and for the USA they retrieved 430 results. After the first round of sifting by two reviewers 46 potentially eligible websites were identified. Of these 19 websites were later excluded, 13 of which only sold in quantities greater than one or to laboratories/hospitals/workplaces, five who incorporated contact with a health professional before the sale, and one which was withdrawn from sale between the search and extraction. We identified 23 molecular virus testing services[14–36] and 18 antibody testing services[14–16 18 19 21 25 26 28 29 31–34 37–40] meeting the inclusion criteria, sold via 27 websites (25 from the UK[14–34 37–40] and two from the USA.[35 36] One website[40] did not appear in the main search, but was mentioned in many UK news articles, so was included in the cohort. Only two websites using home sampling were identified in the USA, the first and second to be approved by the FDA for this use.[35 36] Basic characteristics of the websites and tests are given in online supplemental table 1.

The websites consisted of 13 private health clinics,[14–17 20 21 24–26 29 36 38 39] four pharmacies,[30 32 34 40] four suppliers of a range of direct to consumer testing online,[18 22 31 37] three laboratories,[23 33 35] two online sexual health specialists[19 27], and one supplier of beauty treatments.[28] All 23 molecular virus tests were laboratory-based tests with home sampling. Of the 18 antibody tests, 17 were laboratory-based tests with home self-sampling, and one was a point-of-care test.[38] The test manufacturer was identifiable for 9/41 (22%) tests, and further details are provided in online supplemental figure 1.

The mean cost of molecular virus testing was £168 (range £65 to £279) in the UK and $135 (range $119 to $150) in the USA. The mean cost of antibody tests was £87 (range £55 to £130) in the UK.

The proportion of websites which met each of the criteria for clear communication (outlined in table 1) is shown in figure 1, and examples of reporting are given in table 2.

### Explaining which test and when to test

All 27 websites stated whether the 41 tests for sale were molecular virus tests or antibody tests, of which 40/41 described the test clearly. Guidance on timing of taking the molecular virus tests and the antibody tests was provided by 15/23 (65%)[15 17–21 25–27 29–31 34–36] and 16/18 (89%)[15 16 18 19 21 25 26 28 29 31 33 34 37–40] websites, respectively. Recommendations on timing and variation in timing of sampling are detailed in figure 2, with several contrary to current advice or opinion.[4]

### Test accuracy and interpretation

Of the 41 tests for sale, the websites reported a measure of test accuracy (sensitivity, specificity, positive or negative predictive value) for 27 (66%) tests: 16/18 (89%) for antibody tests[14 15 18 19 21 25 26 28 29 31 33 34 37–40] and 11/23 (48%) for molecular tests.[14 17 20–22 25 26 33–36] An additional 10/41 (24%) tests (two antibody[16 32] and eight molecular tests[15 16 18 19 23 27 29 32] only reported test performance using unclear terms such as 'accuracy' or 'reliability', for example *"This test has a 99.9% accuracy"*[19] and *"This test offers 99.9% reliability."*[29] Tests with unclear performance values may be referring to analytical performance, such as *"Our test is sensitive to fewer than 100 copies of the target viral RNA, making it a highly accurate test."*[32] For two (5%) molecular tests, no text or values referring to accuracy were reported on the websites.[24 31]

Sensitivity and specificity were the most commonly reported accuracy measures, provided for 27/41 (66%)[14 15 17–22 25 26 28 29 31 33–40] and 22/41 (54%)[15 17 19–22 25 26 28 29 31 33–37 39 40] tests, respectively. Sensitivity estimates ranged from 95% to 100% for antibody tests (n=16)[14 15 18 19 21 25 26 28 29 31 33 34 37–40] and 97.5% to 100% for molecular tests (n=11).[14 17 20–22 25 26 33–36] Specificity estimates ranged from 97.5% to 100% for antibody tests (n=13)[15 19 21 25 26 28 29 31 33 34 37 39 40] and were reported as 100% for all molecular tests (n=9).[17 20–22 25 26 34–36] Five of the 41 tests (13%; two antibody tests[28 31] and three molecular[14 20 33] tests) provided an estimate or statements of sensitivity and/or specificity under conditions of perfect use rather than pragmatic use, for example *"If there are any coronavirus on your swab it will definitely find it."*[33]

**Table 1** Predefined information items which we would expect to be communicated to a person considering purchasing a test for COVID-19, and misinformation items which we would consider inappropriate to communicate, with rationale

| Information item | Rationale |
|---|---|
| **Who should take the test?** | |
| 1. Does the website clearly explain whether it is a test for antibodies (whether you have previously had the disease) or active virus (whether you have it now)? | To help the potential purchaser select the most appropriate test type. |
| 2. Does the website explain when to test? | Accuracy is heavily dependent on timing. Antibody tests undertaken too early have low sensitivity (they make false negative errors, that is, miss cases of COVID-19). Molecular virus tests undertaken very early or too late have reduced sensitivity. |
| **Test accuracy information** | |
| 3. Can you identify the test which is used? that is, the manufacturer | There has been significant media coverage of the accuracy of different manufacturers' tests. Providing this information enables those interested to find out more. |
| 4. Does the website give accuracy to detect cases? (sensitivity) | An informed potential purchaser would want to ensure tests successfully identify COVID-19. |
| 5. Does the website give accuracy to detect non-cases? (specificity) | An informed potential purchaser would want to ensure tests did not misidentify COVID-19. |
| 6. Does the website state how many samples the accuracy claims are based on? | Accuracy data based on few samples is less reliable. While few people may be interested in the detail of the test accuracy study design, the number of samples/patients may be of interest. |
| 7. Does the website give information on the post-test probability of having or ruling out the disease? (Positive predictive value or negative predictive value at any prevalence) | This is the most important accuracy information for a person considering buying a test. For an individual whose molecular virus test result is positive, the positive predictive value gives them the probability that they currently have COVID-19. For an individual whose molecular virus test is negative, the negative predictive value is the probability that they do not currently have COVID-19. For an individual whose antibody test is positive, the positive predictive value is the probability that they have COVID-19 antibodies. For an individual whose antibody test is negative the negative predictive value is the probability that they do not have COVID-19 antibodies. These metrics are dependent on disease prevalence as well as sensitivity and specificity, but can reasonably be calculated with informed estimates of prevalence. |
| 8. Does the website give a link or reference to a journal article of test accuracy? | Indicating the source of these data would help substantiate the claims, and allow interested people to find out more. |
| **Avoiding misinformation about interpreting the test** | |
| 9. Molecular virus test – does the website avoid the inaccurate statement that if you test negative you are not infectious or do not need to self-isolate? | The molecular virus tests are not very sensitive and so negative results may be false negatives, so the individual may still have the virus and be contagious. |
| 10. Antibody test – does the website avoid the inaccurate statement that we know that test positive infers immunity or allows you to put yourself at greater risk of virus exposure? | A positive antibody test could be a false positive, meaning the individual does not have antibodies. Even if it is a true positive we do not know whether the presence of antibodies infers immunity, and how that changes over time as antibody levels drop. |
| **Providing accurate information about interpreting the test** | |
| 11. Molecular virus test – does the website state that if you test positive you should self-isolate? | Individuals who test positive on a molecular virus test are likely to have active virus, and are likely to be contagious. |
| 12. Molecular virus test – does the website state that if you test negative you may still have the disease? | Same rationale as item 9 above, but here we assessed whether the websites gave correct information (in addition to avoiding misinformation). |
| 13. Antibody test – does the website explain that we do not know whether a positive test infers immunity, and/or that you shouldn't put yourself at more risk of exposure if you test positive? | Same rationale as item 10 above, but here we assessed whether the websites gave correct information (in addition to avoiding misinformation). |

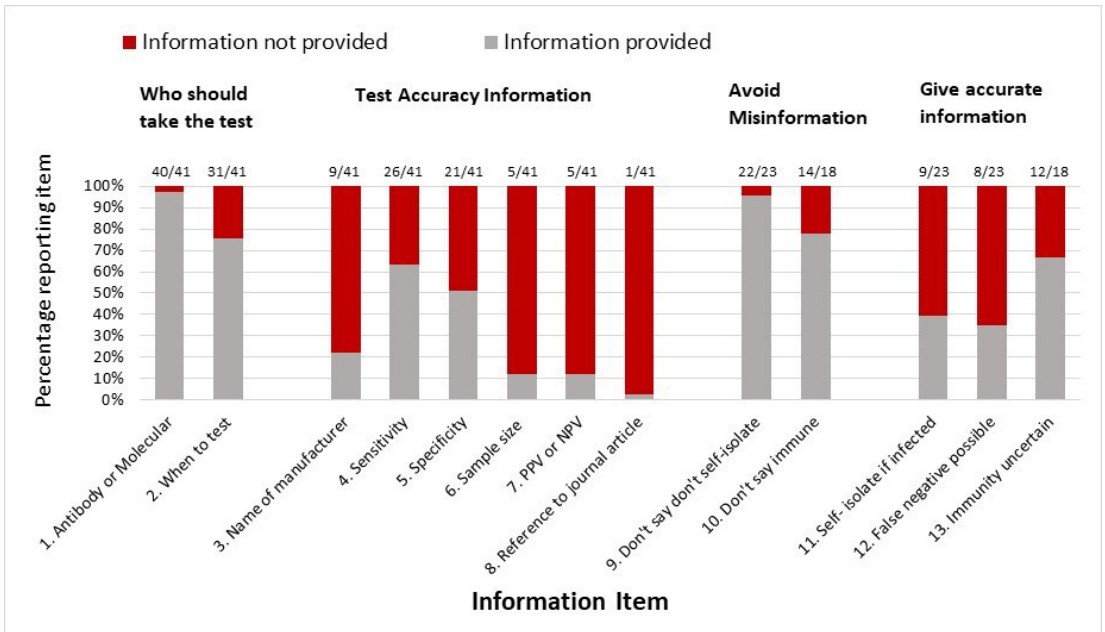

**Figure 1** Proportion of home-sampling COVID-19 tests identified which met/did not meet each of the predefined criteria for clear communication.

No websites directly referred to positive predictive values (PPV), but they were indirectly reported for 5/41 (12%) tests.[20 21 25 25 40] Two antibody[25 40] and three molecular tests[20 21 25] made a statement about the lack of false positives (implying a PPV of 100%), for example *"if it shows a positive result, it can only be for COVID-19".*[25] No cross-reactivity (meaning the test would not identify other viruses, only COVID-19 virus) was referred to by websites for 13/41 (32%) tests (five antibody[16 25 33 34 40] and eight molecular tests).[17 20–22 25 26 28 34] Negative predictive value (NPV) was not referred to by any websites, however, statements implying that the NPV was less than 100% were given for 4/41 (10%) available tests (two antibody[25 31] and two molecular tests,[20 35]) for example *"The test can sometimes show a negative result even if you are infected [with] SARS-CoV-2, the virus that causes COVID-19."*[35]

The number of samples used to generate accuracy data were given for 5/41 (12%) tests: two antibody tests[31 33] and three molecular tests.[22 35 36] Accuracy data were linked to a journal publication for only 1/41 (2%) tests.[33]

Information on interpreting both positive and negative molecular virus test results was presented for 4/23 (17%) websites.[20 33–35] Twelve of the 18 (67%) websites selling antibody tests informed potential customers prior to purchase that a positive antibody test may not infer immunity from future infection[14 16 18 21 25 28 32–34 37 39 40] (figure 1).

Where tests could be identified, we checked accuracy claims against data from published papers, pre-prints (based on information obtained from searches from ongoing Cochrane reviews for these tests), and manufacturer's data in the IFU sheet for each test (table 3). Four websites reported clinical performance data for the Abbott IgG antibody test: two[31 33] quoted the performance figures

from the IFU,[41] for the other two[26 29] no exact match with available studies could be made. Of the four molecular tests, no performance data were available for the Randox test[23] (including in the IFU,[42] no direct match of clinical performance results could be made for the website selling the Primerdesign genesig PCR assay),[22] where the IFU only reported data from contrived samples,[43] whereas the data reported by US websites[35 36] selling the LabCorp and Rutgers PCR tests, respectively, matched data from the manufacturers' IFUs.[44 45]

## Claims about regulatory approval and endorsement

Across the 25 UK websites, there were 17 antibody tests for home sampling,[14–16 18 19 21 25 26 28 29 31–34 37 39 40] one antibody test for home testing[38] and 21 molecular tests for home sampling[14–34] for sale. There was no mention of regulatory approval or endorsement for 18/39 (46%) tests, seven antibody tests[14 16 26 28 32 39 40] and 11 molecular tests[14 16 18 19 22–24 27 29–31] (see online supplemental table 1).

For home sampling antibody tests, 7/17 (41%) included a statement that the test had a CE mark[15 19 21 25 33 34 37] and 7/17 (41%) websites included a clear statement that the test had endorsement from a policy making body such as PHE, the NHS, or the UK or other European governments.[15 18 21 29 31 33 34] This is despite the fact that currently no COVID-19 antibody tests have regulatory approval for home sampling or home testing. Claims being made about home sampling tests were based on approved test use by health professionals using venous rather than finger-prick samples:

"All of our home test kits are CE-marked. This is one of the two IgG tests approved by the Government for UK use."[15]

**Table 2** Examples of clear/accurate and unclear/potentially misleading website content

| Information item | Example of unclear/potentially misleading information | Example of clearer and more accurate communication |
|---|---|---|
| Who should take the test? | *"you can do the swab test between 1–5 days post exposure."*[34]<br>This is likely to be too early for the PCR molecular virus test specified to be sufficiently sensitive. Median time between exposure and symptom onset is around 5 days,[54] so this proposed timing is likely pre-symptomatic when sensitivity is lower. | *"Ideally samples should be taken from symptomatic individuals between days 1–5 from symptom onset. However, there are many cases when virus can be detected later into the illness."*[20]<br>It would also be helpful to communicate that taking the test too early or late when it is less accurate may result in the test missing COVID-19 when it is present. |
| Test accuracy | *"This test offers 99.9% reliability"*[29]<br>*"What is the accuracy of the test? 99.9%"*[29]<br>It is unclear what the terms 'accuracy' or 'reliability' mean. | No website provided a full explanation of accuracy, we suggest our own example as follows (data provided for example and text can be amended to clearly indicate molecular or antibody tests)<br>*"Test accuracy: The tests are sometimes inaccurate. If you have a negative result (indicating you have not got COVID-19) then the test is very likely to be correct. If you get a positive result (indicating you have got COVID-19) then the result is less accurate. Of the people who test positive, 92 in 100 do actually have COVID-19. Of the people who test negative, more than 99 in 100 do not have COVID-19. Here is more detail on the science: Test accuracy was measured in an independent evaluation of 158 people with COVID-19 and 364 people without COVID-19 (give reference for the underlying evidence). The test had a sensitivity of 98% and a specificity of 99.2%. That means that if 1000 people are tested, and 100 of those have COVID-19, then 98 of the 100 people with COVID-19 will be detected and two will be missed (test negative). Of the 900 people who do not have COVID-19, 892 will test negative, and eight will test positive (and believe they have COVID-19 when they do not)."* |
| Interpreting test results of molecular virus tests | *"This highly accurate test will give you peace of mind that you can't infect others. This test is relevant when people who have been isolating wish to return to their household, community or workplace and need to know that they aren't infectious"*[16]<br>This refers to the PCR molecular virus test, which is known to have low sensitivity so people testing negative may still be infected and infectious to others. Reasons for taking the PCR test cited as *"You need to know if you are infectious or not"* and *"You want to let your household members know if they need to self-isolate"*[25] | *"If you have tested positive for COVID-19, self-isolation is recommended so that you do not pass the virus to others…If your results are negative and you're having symptoms, continue to follow isolation precautions and ask your healthcare provider if you need further testing."*[35]<br>Linking information on the low negative predictive value of the PCR test to recommendations to continue self-isolation may strengthen the message. |
| Interpreting test results of antibody tests | *"A positive test result indicates that you have been exposed to COVID-19 and your immune system has produced antibodies in response to the virus. If you have had no symptoms for at least 7 days, you should have some level of immunity to COVID-19 and may not be able to transmit the virus to others or become infected by it again."*[19]<br>We do not currently know whether the presence of antibodies infers immunity. | *"There is still a great deal about COVID-19 immunity that we do not yet fully understand… If your IgG test is positive it means you have had COVID-19 exposure sufficiently to make an antibody response to the virus. There is currently no scientific evidence confirming if the presence of antibodies correlates to immunity or how long the antibodies will last for."*[21] |

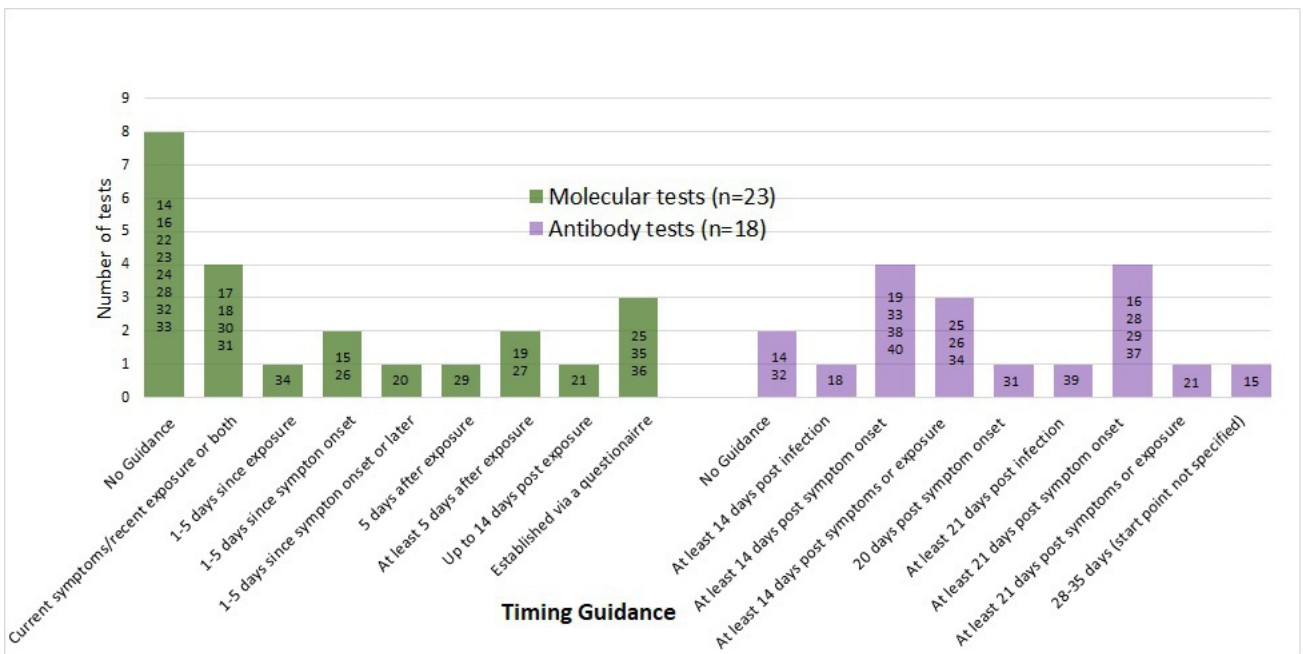

**Figure 2** Recommendations given by websites on when to take the molecular virus tests and antibody tests. Test accuracy is dependent on correct timing.

One website[38] claimed it had regulatory approval for its home testing antibody test:

"Our test has been accepted by Medicines and Healthcare products Regulatory Agency (MHRA), which means that it can be applied across the EU including UK. We confirm our product can meet the requirement of in vitro Diagnostic Medical Devices Directive (98/79EC) and standards complying with CE Declaration of Conformity."[38]

Five of 21 (24%) UK websites selling molecular virus tests for home sampling included a statement that they had regulatory approval,[15 21 25 26 33] and six (29%) websites[17 20 28 32–34] claimed approval from a policy making body for this intended use. The manufacturer or name of the molecular tests for which websites were claiming regulatory approval or endorsement could not be identified. Only for two websites selling molecular tests,[22 23] the test manufacturer could be identified, neither of which made any claims about regulation or endorsement. One of these tests[23] is mentioned by the UK government as part of its COVID-19 testing strategy[46] and the other[22] was one of the tests which was independently evaluated by PHE.[47]

Both US websites selling molecular viral tests[35 36] have approval from the FDA for home sampling during the COVID-19 pandemic. These websites included information about the eligibility checks that purchasers would need to undergo either prior to purchase or prior to test processing.

We reviewed the 18 UK websites selling home COVID-19 antibody tests[14–16 18 19 21 25 26 28 29 31–34 37–40] on 11/12 June 2020 after the MHRA had instructed sales of these tests to cease because of the lack of approval for the tests using finger-pick samples.[11] We found two websites[32 38] that appeared to still be selling finger-prick tests, four[14 21 28 31] had switched to providing a venous blood sampling service, two[18 33] required the purchaser to find their own phlebotomist to draw a blood sample to send, six[15 16 19 25 34 40] simply stated that tests were out of stock and were unavailable, while four[26 29 37 39] reported the MHRA guidance and indicated that they had suspended sales (table 4).

## DISCUSSION

We identified 27 websites selling COVID-19 tests direct to the public, 25 in the UK but only two in the USA, which may be explained by the FDA stipulations requiring clinician involvement in the testing process. We observed that many websites failed to provide complete information on the name and manufacturer of the test (no information for 32/41 tests), when to use the test (no information for 10/41 tests), the accuracy of the test (no information for 12/41 tests), and how to interpret results (no information for 21/41 tests), which will hinder the public making informed choices about testing, using tests correctly and understanding what test results mean. Without adequate and correct information the public may purchase the wrong or a poor test, or use the test in the wrong way or at the wrong point in time. These errors or applications will increase their chances of getting an erroneous test result. Even when used properly, few websites assisted users in interpreting test results and understanding their inherent uncertainty.

This rapid evaluation was designed to provide timely results in the context of a fast-moving global pandemic. The search was not designed to be exhaustive, rather to represent what a person typing "coronavirus test" or similar into a Google search would have retrieved. Using

**Table 3** Claims of test accuracy from websites (selected verbatim) and evidence identified from the manufacturers' instructions for use sheet (IFU), published papers and pre-prints

| Website information | | | Published information | | |
|---|---|---|---|---|---|
| **What do they say about accuracy?** | **Sensitivity (%)** | **Specificity (%)** | **Study details** | **Sensitivity (%)** | **Specificity (%)** |
| **Abbott's antibody test (Abbott Architect SARS-CoV-2 IgG)** | | | | | |
| **London Medical Laboratory**[33] **Sensitivity:** This test has proven to be 100% accurate in identifying antibodies to SARS-CoV-2 coronavirus at 14 days after onset of COVID-19 symptoms. **Specificity:** It is 99.63% specific. Or put another way, only 0.37% of over 1000 people tested who could not have been exposed to SARS-CoV-2 showed a false positive result. | 100 (n=88) | 99.63 (n=1070) | **Manufacturer's clinical performance of test**[41] ▶ **Positive samples tested:** 122 serum and plasma specimens were collected at different times from 31 subjects who tested positive for SARS-CoV-2 by a polymerase chain reaction (PCR) method and who also presented with COVID-19 symptoms. ▶ **Negative samples tested:** 1070 specimens, 997 specimens were collected prior to September 2019 (pre-COVID-19 outbreak). An additional 73 specimens were collected in 2020 from subjects who were exhibiting signs of respiratory illness but tested negative for SARS-CoV-2 by a PCR method (unclear how many participants) ▶ **Reference standard:** PCR method | 100 (n=88) | 99.63 (n=1070) |
| **MyHealthcare Clinic**[29] The manufacturer of the antibody test (Abbott Laboratories) reports that an independent clinical performance evaluation of the test performed in the UK confirmed the following accuracy levels: **Sensitivity** (ability to identify positive cases) of 99.7%. **Specificity** (ability to identify negative cases) of 100%. | 99.7 (n=NR) | 100 (n=NR) | **Bryan et al 2020**[55] ▶ **Positive samples tested:** 125 patients who tested RT-PCR positive for SARS-CoV-2 for which 689 excess serum specimens were available (unclear how many at each time point). ▶ **Negative samples tested:** 1020 serum specimens collected prior to SARS-CoV-2 circulation in the USA ▶ **Reference standard:** PCR method and pre-COVID-19 samples | 53.1 (at 7 days) (n=NR) 82.4 (at 10 days) (n=NR) 96.9 (at 14 days) (n=NR) **100.0 (at 17 days—data driven choice) (n=NR)** | 99.9 (n=1020) |
| **The Online Clinic (Online Clinic (UK) Limited)**[26] Tests have a **sensitivity** of 100%. When using a patient-collected sample with one of our home sampling kits, the sensitivity of this test has been shown to reduce slightly to 97.5. Recent studies suggest a **specificity** of 99.9% and 99.8% respectively. | 97.5–100 (n=NR) | 99.8–99.9 (n=NR) | **Phipps et al 2020**[56] ▶ **Positive samples tested:** Only six patients with samples 14 days post-symptom onset, the point at which the highest sensitivity was recorded. 173 suspected COVID-19 cases with 76 were confirmed positive by PCR methods. ▶ **Negative samples tested:** Plasma samples from healthy donors (2019 blood donations and 2020 blood donations from healthy donors without recent illness) ▶ **Reference standard:** PCR method for suspected COVID-19 cases to confirm positives; for negatives apparent healthy donors | 38 (all days) (n=76) 7 (<3 days) (n=15) 30 (3–7 days) (n=27) 33 (5–15 days) (n=15) **83 (after 14 days) (n=6)** | 100 (n=656) |

Continued

**Table 3** Continued

| Website information | | | Published information | | |
|---|---|---|---|---|---|
| What do they say about accuracy? | Sensitivity (%) | Specificity (%) | Study details | Sensitivity (%) | Specificity (%) |
| **Atruchecks Limited**[31] Abbott claim their test is 100% **sensitive** (88 samples) and 99.6% **specific** (1070 samples). Our UK Accredited Partner Lab have run their own internal verification of these claims and achieved a **sensitivity** of 98.5% (132 samples) and a **specificity** of 99.5%. (186 samples). | 98.5 (n=132) 100 (n=88) | 99.5 (n=186) 99.6 (n=1070) | **Public Health England (PHE) evaluation of the Abbott antibody test**[57] ▶ **Positive samples tested:** 96 samples defined by a positive PCR from a swab sample for that patient ▶ **Negative samples tested:** 759 negative samples were included in the evaluation (351 samples that are rheumatoid factor, CMV, EBV, or VZV positive; 11 seasonal coronavirus positive samples; and 395 historic samples, two samples unclear). These samples have been chosen based on their collection before mid-2019 to ensure they are SARS-CoV-2 antibody negative, but will contain samples containing antibodies to other seasonal coronaviruses to provide an additional screen for the assay ▶ **Reference standard:** PCR method and pre-COVID-19 samples | 92.7 (all days) (n=96) 93.4 (≥14 days) (n=82) 93.9 (≥21 days) (n=76) | 100 (n=759) |

**Epitope Diagnostics inc. (EDI) Antibody test (EDI Novel Coronavirus COVID-19 IgG ELISA Kit)**

| | | | | | |
|---|---|---|---|---|---|
| **Summerfield Healthcare**[16] We have a trusted product which is **specific** to COVID-19 and **sensitive**. As with all of these kits they undergo regular testing to ensure **accuracy** and **reliability** which on the last occasion were **100% accurate for both positive and negative samples.** Antibody test is very specific for COVID-19 (some inferior tests can mistake other infections for COVID-19 and wrongly reassure you). It is also very sensitive for the specific IgG antibody. | NR (n=NR) | NR (n=NR) | **Manufacturer's clinical performance of test**[58] ▶ **Positive samples tested:** RT-PCR confirmed positive patients. ▶ **Negative samples tested:** Normal healthy patients with samples collected prior to the COVID-19 outbreak. ▶ **Reference standard:** PCR method and pre-COVID-19 samples | 98.4 (n=187) | 99.8 (n=624) |
| | | | **Krüttgen et al 2020**[59] ▶ **Positive samples tested:** The sera of the 31 patients with positive SARS-CoV-2 PCR were collected 11.9 days (±5.0 days) post-onset of symptoms. 22 sera were considered SARS-CoV-2 IgG positive (positive on at least two assays). ▶ **Negative samples tested:** 53 sera were regarded as IgG negative ▶ **Reference standard:** A serum was regarded as SARS-CoV-2 IgG negative if at least three of the four assays compared here (for the Euroimmun assay, the EDI assay, the Mikrogen assay, and the Viramed assay) had a negative test result applying the manufacturer's interpretation criteria. A serum was regarded as SARS-CoV-2 IgG positive if at least two of the four assays had a positive test result (comparator tests are also part of reference standard). | 100 (n=22) | 88.7 (n=53) |

**Randox PCR antigen test (Randox COVID-19 Home Testing Kit)**

**Table 3** Continued

| Website information | | | Published information | | |
|---|---|---|---|---|---|
| What do they say about accuracy? | Sensitivity (%) | Specificity (%) | Study details | Sensitivity (%) | Specificity (%) |
| **Randox**[23]<br>This is a PCR-based test, utilising Randox Biochip Technology, to provide an **accurate diagnosis** for COVID-19. | NR<br>(n=NR) | NR<br>(n=NR) | **Manufacturer's clinical performance of test**[42]<br>▶ **Positive samples tested:** NR<br>▶ **Negative samples tested:** NR<br>▶ **Reference standard:** NR | NR<br>(n=NR) | NR<br>(n=NR) |
| | | | **Public Health England (PHE) evaluation of the Randox antigen test**[60]<br>▶ **Positive samples tested:** None<br>▶ **Negative samples tested:** The assessment sample-panel totalled 195 specimens, including upper or lower respiratory clinical specimens negative for SARS-CoV-2 as determined by the validated in-house PHE PCR assay and dilutions of SARS-CoV-2<br>▶ **Reference standard:** PHE PCR assay | NR<br>(n=0) | 100<br>(n=195) |

**Primerdesign Ltd antigen test (Coronavirus (COVID-19) genesig Real-Time PCR assay)**

| Website information | | | Published information | | |
|---|---|---|---|---|---|
| What do they say about accuracy? | Sensitivity (%) | Specificity (%) | Study details | Sensitivity (%) | Specificity (%) |
| **Rightangled Healthcare**[22]<br>Studies confirm Primerdesign COVID-19 assays are **highly specific** for the detection of SARS-CoV-2 virus and detection of coronavirus COVID-19 disease.<br>Independent Clinical Performance Evaluation of Primerdesign COVID-19 assay by the National Infection Service, Public Health England, Colindale confirmed the **specificity** of this assay using upper or lower respiratory clinical samples from patients and known SARS-CoV-2 positive material. PHE confirmed the assay showed >98% specificity to SARS-CoV-2 virus in clinical samples.<br>An Independent Clinical Performance Evaluation by an NHS Clinical Pathology Laboratory using patient samples with respiratory symptoms confirmed the assay was 100% **specific** when tested against known positive and negative SARS-CoV-2 clinical samples. | 98<br>(n=50) | 100<br>(n=50) | **Manufacturer's clinical performance of test**[43]<br>▶ **Positive samples tested:** Contrived oropharyngeal swabs (50 positive)<br>▶ **Negative samples tested:** Contrived oropharyngeal swabs (50 negative)<br>▶ **Reference standard:** 50 swabs were contrived with SARS-CoV-2 whole viral genomic RNA | 94.7 (1–2x LoD)<br>(n=38)<br>100 (3x LoD)<br>(n=7)<br>100 (4–5x LoD)<br>(n=5) | 100<br>(n=50) |
| | | | **van Kasteren et al 2020**[61]<br>▶ **Positive samples tested:** Clinical samples previously submitted for routine SARS-CoV-2 diagnostics for which the presence of various amounts of SARS-CoV-2 RNA had been confirmed using in-house PCR.<br>▶ **Negative samples tested:** Clinical samples with confirmed respiratory viruses (influenza virus type A (n=2), rhinovirus (n=2), RSV-A and -B)<br>▶ **Reference standard:** SARS-CoV-2 RNA had been confirmed using in-house PCR | 62.5<br>(n=16) | 100<br>(n=6) |
| | | | **Public Health England (PHE) evaluation of Primerdesign antigen test**[47]<br>▶ **Positive samples tested:** None<br>▶ **Negative samples tested:** : The assessment sample-panel totalled 195 specimens, including upper or lower respiratory clinical specimens negative for SARS-CoV-2 as determined by the validated in-house PHE PCR assay, and dilutions of SARS-CoV-2<br>▶ **Reference standard:** PHE PCR assay | NR<br>(n=0) | 100<br>(n=195) |

**Table 3** Continued

| Website information | | | Published information | | |
|---|---|---|---|---|---|
| What do they say about accuracy? | Sensitivity (%) | Specificity (%) | Study details | Sensitivity (%) | Specificity (%) |
| **LabCorp antigen test (COVID-19 RT-PCR test with Pixel by LabCorp COVID-19 test home collection kit)** | | | | | |
| **LabCorp**[35] Customers provided with link to FDA Emergency Use Authorisation Summary. | 100 (NP swabs) (n=40) 100 (BALs) (n=40) | 100 (NP swabs) (n=50) 100 (BALs) (n=50) | **Manufacturer's clinical performance of test**[45] ▲ **Positive samples tested:** Positive samples were comprised of 40NP swabs and 40 BALs spiked with quantitated live SARS-CoV-2. 10 samples each were spiked at 8×, 4×, 2×, and 1× LoD. ▲ **Negative samples tested:** Negative samples include 50NP swabs and 50 BALs. ▲ **Reference standard:** Contrived samples | 100 (NP swabs) (n=40) 100 (BALs) (n=40) | 100 (NP swabs) (n=50) 100 (BALs) (n=50) |
| **Rutgers Clinical Genomics antigen test (Rutgers Clinical Genomics Laboratory TaqPath SARS-CoV-2 Assay)** | | | | | |
| **Hims**[36] Customers provided with link to FDA Emergency Use Authorisation Summary. | 100 (n=30) | 100 (n=30) | **Manufacturer's clinical performance of test**[44] ▲ **Positive samples tested:** 30 contrived positive samples were tested ▲ **Negative samples tested:** 30 contrived negative samples were tested ▲ **Reference standard:** Contrived samples | 100 (n=30) | 100 (n=30) |

different phrases such as "coronavirus antibody test" would have identified additional websites, but there is no reason to suspect that they would be different from those summarised here. The timing of the search and data extraction will have affected results. Data extraction was shortly before the UK MHRA clarified that antibody tests were not approved for finger-prick samples, only for venous samples. The search only identified two US websites selling tests with home sample collection, but at the point of going to press eight tests are now approved on the FDA website.[48] The criteria that we used to assess completeness of communication (detailed in table 1) were defined *a priori*, but due to time constraints a formal process for developing these was not followed. However, all key elements of the search, selection, and data extraction processes were undertaken independently by two researchers, reducing the possibility of errors. We only assessed information provided prior to purchase, as complete information should be given at this stage to inform the purchasing decision. However, further information would have been given after purchase, for example within the instructions for use, which was beyond the scope of this paper.

The issues we have identified are examples of poor and misleading practice, and some merit further investigation by the MHRA and Advertising Standards Authority (ASA). At the time of going to press two antibody tests remained on sale and we have reported these to the MHRA. The communication of test accuracy appears to contravene advertising standards in the UK. The five websites that reported PPV of 100% contrary to the wider evidence base, and all websites making accuracy claims which is not linked to supporting evidence appear to contravene section 12.1 of the ASA code,[49] which states that objective claims must be backed by evidence. Further, websites provided specificity and sensitivity, or general claims of 'accuracy' rather than positive and negative predictive values explained in lay terms, and the ASA have previously ruled against this practice as misleading in the case of non-invasive prenatal testing for trisomies.[50] Finally, the lack of complete information on the implications of positive and negative test results does not appear to be covered by any UK regulation, perhaps because the ASA 12.2.1[49] prohibits diagnosis by post or email, and so this information is intended to be provided by contact with a healthcare professional. While such contact with a health professional is happening in the USA it does not appear to be in the UK. Regulation of product labelling provides a means to oversee information communicated for self-testing products bought in person, but there is currently no equivalent for online testing services in the UK. This gap in regulation could be solved by expanding the responsibility of the MHRA to include communication by 'distributors' at the point of online purchase, working collaboratively with the ASA. There was a large variation in the price of testing in the UK, and in many cases these differences do not appear to be justified by differences in the service provided. Greater regulation and

**Table 4** Availability of finger-prick antibody tests post-MHRA withdrawal from market notice (websites accessed on 11 to 12 June 2020)

| Website | Status | Comments |
|---|---|---|
| PillDoctor[32] | Still available | *Test appears to still be available for purchase* |
| YourHealthFirst Clinic[38] | Still available | *Test appears to still be available for purchase* |
| Summerfield Healthcare[16] | Not currently available | *Webpage suggests finger-prick antibody test still available but not available in subsequent drop down menu.* |
| Doctorcall[15] | Not currently available | "Coming soon"<br>*Option on website to be notified when product is back in stock* |
| WebMed Pharmacy[34] | Not currently available | "Sorry the item you have selected is not currently available, please choose another option"<br>"Due to the high demand of orders, the antibodies blood test service is currently not available." |
| Superdrug[40] | Not currently available | "We have temporarily halted the COVID-19 antibody testing service. If you have any questions please send us a message through your account." |
| Zava[25] | Out of stock | "This product is temporarily out of stock."<br>*Option on website to be notified when product is back in stock* |
| Better2Know[19] | Out of stock | "Currently out of stock"<br>*Website links to guidance from MHRA.* |
| Antibody Solutions[28] | Out of stock/ modified test | "Please note: these kits are no longer in stock; however, we are offering a full blood sample collection service, either at your home or at one of our partner clinics." |
| Blue Horizon Medicals[18] | Modified test | "Ordering this test will allow us to send you a vacutainer kit, which allows a healthcare professional to draw a venous blood sample from your arm. You should only order this kit therefore if you have access to a healthcare professional with the appropriate skills. Phlebotomy should NOT be attempted by those who are unskilled." |
| Qured[21] | Modified test | "A healthcare professional will visit your home to take a venous blood sample."<br>"The antibody tests currently used by our laboratory are the Abbott test if you opt for venous blood collection by a healthcare professional, or the Siemens test if you opt to collect your blood sample yourself"<br>"These tests are currently validated for venous blood draw only, which is why our service includes an at-home blood draw from a healthcare professional. Home self-collection of blood using a finger-prick kit for antibody testing has been temporarily paused pending evaluation by the Medicines and Healthcare products Regulatory Agency (MHRA)." |
| CityDoc[14] | Modified test | "We are able to offer blood collection by the normal practice of intravenous blood sampling at our clinics across the UK and sent to our accredited UK laboratory for testing." |
| Atruchecks Limited[31] | Modified test | "PHE approved Abbott test in our accredited lab. Venous sample taken in central London clinic, off Harley Street (W1)." |
| London Medical Laboratory[33] | Modified test | "This option is so you can arrange a home or workplace visit by a phlebotomist to take your blood for you." |
| The Online Clinic (Online Clinic (UK) Limited)[26] | Suspended/ modified test | "The self-collect home sampling service is currently suspended but will be back shortly. Please check back later."<br>"The Medicines and Healthcare Regulatory Agency is currently conducting a review of self-collect blood samples for this type of test and the service is unavailable until that review concludes. We now offer a home-sampling service where a phlebotomist attends your home (or other premises) to collect the blood sample from a vein. " |

Continued

**Table 4** Continued

| Website | Status | Comments |
|---|---|---|
| MyHealthcare Clinic[29] | Withdrawn/ suspended | "We have unfortunately had to withdraw the PHE Approved Antibody Home Testing Kits, per the unexpected Government / MHRA ruling on 26 May re private testing. We do not currently have a date for when these Home Tests will be next available to private patients." |
| Medichecks[37] | Withdrawn/ suspended | "Currently, the only way to get a private coronavirus antibody test is to buy a venous blood test where you will need to visit a nurse or health professional to have a sample collected from a vein in your arm. All private laboratories and private testing companies have paused self-collect finger-prick testing while the MHRA conducts its review. However, we are confident that this service will resume shortly once the laboratories have completed their validation studies." |
| Babylon[39] | Withdrawn/ suspended | "Important update on COVID-19 Antibody Tests. The MHRA (the government regulator responsible for medicines and medical devices) has asked that all COVID-19 antibody testing from finger-prick blood samples be paused. The MHRA decision has impacted all testing of this type nationwide." "The lab will not be offering further testing services until the MHRA have provided clearance to do so." |

standardisation of website claims may reduce this price differential by making comparisons between websites easier, and removing unsubstantiated claims.

### Key communication requirements

It is important that all test users are given adequate and appropriate information to help them make safe and informed choices. We identified five key communication issues with websites selling direct to consumer home-sampling COVID-19 tests. All five of these issues may be improved by developing a basic framework of what information should be provided, and standard ways to present such information. This would also facilitate comparison between websites.

### The type of test and the questions which it can help address

It is essential that companies selling tests identify the type of test, and the situations in which it is appropriate to order such a test. While websites were clear whether they were selling molecular or antibody tests, they also need to indicate the situations when it is appropriate to order a molecular "swab" test or an antibody "blood" test in order to select the correct one. The two US websites used questionnaires recording symptoms and exposure which were reviewed by clinicians prior to tests being despatched, which provides a more rigorous check on whether the test request is sensible.

### How and when the test should be used

Both molecular and antibody tests need to be used at different time points in the disease course. The sensitivity of both types of tests will fall if used at the wrong time point (sensitivity of 31% for antibody tests in the first week since onset of symptoms,[3] substantially increasing the risks of infection or antibody response being undetected).

Recommended time points when samples should be taken were absent for 10/39 UK tests (26%). Some timing statements were misleading, suggesting using the test at time points which are known to be too early or too late. Some websites stated dates based on time since exposure, others since symptom onset which is median 5 days after exposure. Both are required to be able to advise both asymptomatic patients and patients with unknown exposure when they should order and use the tests.

Websites must also describe the full testing process and clearly indicate what is required of users to complete testing. For example, two antibody websites currently indicate that purchasers will need to identify individuals qualified to take venous blood samples, which is impractical for most people.

### The test name, evidence of its accuracy, and evidence of its regulatory approval for the purpose to which it is put

The majority of tests were for sale by third parties, ranging from healthcare providers to beauty treatment specialists. In most cases (32/41; 78%) it was not possible to identify the test being used or the manufacturer. This does not allow the individual to know the product that they are buying, and precludes the opportunity for the user to verify its regulatory status and the claims being made.

Information on test accuracy was absent or uninterpretable for 12/41 (29%) tests. Numerical accuracy claims could only be matched to published evidence for 4/41 (10%) of tests. In these instances, figures most closely matched those from the manufacturers' IFU leaflets, which tended to report the highest observed values of sensitivity and specificity, and were based on studies more akin to analytic validity than clinical validity evaluations. Accuracy measures from analytic validity studies

should not be assumed to give a good representation of test accuracy when applied in practice to the public. A wide range of terms were used, several of which did not have a clear meaning. It appeared that test accuracy data is not available at all for some tests (Randox),[42] or only based on contrived samples (Primerdesign)[43] and not on real patients. For molecular COVID-19 tests, no clinical performance data were available that were based on self-sampled swab tests. Withholding the fact that there is no patient-based evidence of the accuracy of these tests from the public is unacceptable. It is important that the reported accuracy is based on all reviewed evidence and not selected results, and clearly explains how applicable the evidence is to the public.

Naming tests is essential to be able to check their regulatory status. Seven of 18 (39%) UK websites selling antibody tests inappropriately claimed CE marking, when the CE marking was for a different intended use (venous rather than finger-prick blood samples). Antibody tests are not approved for home use in the USA, and none were found in our search. The UK regulator acted after we had reviewed UK websites, clarifying that antibody tests which are approved for the use with venous samples should not be sold for the use with finger-prick samples. However, 2/18 remain available for online sale at the point of going to press (accessed 11 to 12 June 2020). The molecular tests we could identify are approved for home sampling, however, the name and manufacturer was not identifiable for most websites.

### What test results mean

Research concerning the communication of test accuracy evidence is limited and is largely restricted to self-selected, professional, and postgraduate student groups.[51] Communication of test accuracy evidence is complex for several reasons. Research has highlighted the importance of communicating the potential consequences of positive and negative test results (use of predictive values) and the importance of contextualising estimates of accuracy with reference to a healthcare setting (for example, hospital in patient, hospital outpatient, community).[52 53] Presenting test accuracy as frequencies rather than as probabilities improves understanding.

To interpret results, test users need to know how to interpret positive and negative test results (predictive values), not the proportion of cases detected (sensitivity) and non-cases correctly diagnosed (specificity). Positive predictive value was only reported for 5/41 (12%) tests, and in all five they claimed it was 100% which is inconsistent with the broader evidence base. Negative predictive value was not reported at all.

Most websites gave insufficient information regarding the interpretation of test results. Only 8/23 (35%) websites explained that a negative molecular virus test does not rule out COVID-19, and only 12/18 (67%) explained that a positive antibody test does not necessarily infer immunity from future COVID-19 infection or transmission.

### Decisions which could be made based on the test results

Misunderstanding of the implications of test results could mean that individuals put themselves or others at risk of infection in the mistaken belief that they do not have COVID-19, or that they are immune to COVID-19. This last category probably has the greatest potential for harm. Clear communication about the meaning of test results as detailed above should be linked to evidence-based guidance about behaviour modification in light of test results. We found widespread evidence of websites failing to provide such evidence-based guidance, and some cases of websites actively suggesting unsafe behaviour.

### CONCLUSIONS

At the point of online purchase of home self-sampling COVID-19 tests, users in the UK are provided with incomplete, and, in some cases, misleading information on test application, accuracy, and interpretation. Many websites omit trustworthy guidance on the timing of tests, the interpretation of positive and negative test results, and the implications of results. Best practice guidance for communication about tests to the public should be developed and the role of the regulator in enforcing complete and accurate information should be reviewed. This should be underpinned by robust collaborative qualitative research exploring how members of the public interpret information and measures of accuracy, thus informing how it can be provided in a way that is clear, complete, and accessible.

**Contributors** STP, SB, AJS, KF, MJP, CD, JG, IMH, OO, MS, and JJD contributed to the conception of the work and interpretation of the findings. OO and JG performed the Google searches. STP, SB, KF, JG, OO, IMH, and MJP extracted the data. STP, AJS, MJP, and CD undertook the analysis and drafted the manuscript. STP, SB, AJS, KF, MJP, CD, JG, IMH, OO, MS, and JJD critically revised the manuscript and approved the final version. STP acts as guarantor. The corresponding author attests that all listed authors meet authorship criteria and that no others meeting the criteria have been omitted.

**Funding** This paper presents independent research supported by the NIHR Birmingham Biomedical Research Centre at the University Hospitals Birmingham NHS Foundation Trust and the University of Birmingham. T-P is supported by an NIHR Career Development Fellowship (CDF-2016- 09-018). MS is supported by the National Institute for Health Research (NIHR) Applied Research Collaboration (ARC) West Midlands. KF is funded by the NIHR through a doctoral research fellowship.

**Disclaimer** The views expressed are those of the author(s) and not necessarily those of the NHS, the NIHR, or the Department of Health and Social Care.

**Competing interests** MS reports grants from NIHR Applied Research Collaboration WM; AJS, SB, MP, CD, and JD report funding and support from NIHR Birmingham Biomedical Research Centre.

**Patient consent for publication** Not required.

**Provenance and peer review** Not commissioned; externally peer reviewed.

**Data availability statement** All data relevant to the study are included in the article or uploaded as supplementary information. No additional data available.

of the translations (including but not limited to local regulations, clinical guidelines, terminology, drug names and drug dosages), and is not responsible for any error and/or omissions arising from translation and adaptation or otherwise.

**ORCID iDs**
Sian Taylor-Phillips http://orcid.org/0000-0002-1841-4346
Karoline Freeman http://orcid.org/0000-0002-9963-2918
Isobel M Harris http://orcid.org/0000-0001-8125-3832

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
