## [Reviewer comments · BMJ Open]

This paper was submitted to a another journal from BMJ but declined for publication following peer review. The authors addressed the reviewers' comments and submitted the revised paper to BMJ Open. The paper was subsequently accepted for publication at BMJ Open.

ARTICLE DETAILS

TITLE (PROVISIONAL)	Information given by websites selling home self-sampling COVID-19 tests: An analysis of accuracy and completeness
AUTHORS	Taylor-Phillips, Sian; Berhane, Sarah; Sitch, Alice J; Freeman, Karoline; Price, Malcolm James; Davenport, Clare; Geppert, Julia; Harris, Isobel M; Osokogu, Osemeke; Skrybant, Magdalena; Deeks, Jonathan J

VERSION 1 – REVIEW

REVIEWER	margaret mccartney gp gp
REVIEW RETURNED	26-Jun-2020

GENERAL COMMENTS	page 2 - instead of saying 'websites' should indicate type of test/limitations, would be better to say 'companies' so that better info is recommended to be included in all sales points page 3 ref 2 - This says higher sensitivities https://www.bmj.com/content/369/bmj.m2066 for antibody testing - if disagree helpful to say why page 4 - It would be really helpful to know the proportion of online sales which did make people go through some kind of assessment with video/healthcare professional first. I understand why not included but v useful information to know more broadly ie are these tests in general being sold with no extra information (a follow up would be really useful to find how much better these sets of information are...) Box 1 - am not sure that the example of better communication for 'who should take the test' really is much clearer. It needs framed in days surely - appreciate text abbreviated. problem is that gov information is currently worse ... ideally I think this should say eg 'Testing too early or too late can give a negative test even when you do have the virus. The best time to test is.... however...' etc. may be better to say what points need stated in order to give fair info. Appreciate this is not going to appear very often on sales sites but by explaining what is ideal/needed I think will help ASA in particular
---

	when they review their guidance. second on 'test accuracy' - this relates to antibody testing from the reference and is not a test for current covid-19 as per the example of better communication. needs a top line eg "all tests have some degree of inaccuracy. If you have a negative result (you have not got antibodies to covid-19) then the test is very likely to be correct. If you get a positive result (you have got antibodies to covid-19) then the result is less accurate.' also worth saying that 'accuracy' is a statistical term but ? is it being used in the statistically correct way here ? in the other boxes there are still things I'd want to know as a customer (ie If the tests find covid-19, then it is very accurate. If the test is negative for covid-19, then it is very unreliable. Up to a third of people who do have covid-19 will test negative even though they do have the disease. and top line Antibody tests are not very helpful for most people The only other thing I'd suggest is a box explaining kite marks/ CE marks, what needs MHRA approval , and not - it's very confusing and a clear brief explanation would be useful - this can relate to who needs to do what in order to make it better (am always alarmed at burden regulation ASA - a voluntary org who have to buy in expertise - are relied on for sorting out medical claims)
--	---

REVIEWER	Ian S. Watt Emeritus Professor Hull York Medical School/ Department of Health Sciences, University of York
REVIEW RETURNED	26-Jun-2020

GENERAL COMMENTS	This was a well written manuscript describing a cross-sectional observational study which had assessed the accuracy and completeness of information provided by websites selling home self-sampling and testing kits for COVID-19. Whilst previous studies have considered the accuracy and utility of information on health related websites, to my knowledge this is the first to do so for COVID-19 testing kits. The study findings are of importance for research, health professional, policy and lay audiences. I would support publication of the paper and have only a few minor suggestions to make re clarifying some of the detail in the paper. These are outlined below: * in the first sentence of the final paragraph of the introduction I am unclear as to whether the "home sampling" refers only to molecular virus testing or to antibody testing as well. *I was unclear what a "representative sample of websites" meant in the description of methods * In the results it would be helpful to have more detail on how the number of results in the Google searches (550 in the case of the UK) relates to the number of websites and in turn the number of tests identified * I am unclear as to if any of the tests had been purchased whether
--

	further information might have been forthcoming when the tests were dispatched * In the conclusion the authors mention a number of entirely reasonable recommendations. Do the authors have any suggestions who should be responsible for them - for example who should develop best practice guidance for communication about tests to the public.
--	---

REVIEWER	Hazel Thornton Honorary Visiting Fellow University of Leicester
REVIEW RETURNED	27-Jun-2020

GENERAL COMMENTS	This analysis of accuracy and completeness of user information provided by sellers` websites selling home self-sampling COVID-19 tests is an original and important piece of work that clearly illustrates the pitfalls of a poorly regulated marketplace. It illuminates the urgent need for regulatory reform, ultimately enabling provision of trustworthy guidance to users of websites seeking commercial home testing kits – another small step to achieving more independence from unregulated commercial influences. [1] [2] I suggest this should be emphasized. The manuscript is well organized and generally clearly written. This piece of work will add to the accumulating information being published in the BMJ, both concerning the Covid-19 pandemic and its campaign to tackle commercial influences in health provision. It will be of particular interest to citizens, health professionals and policymakers. It has been methodically researched by a strong team to a pre-defined plan formulated to capture all the information that a purchaser would require in order to determine a test`s benefits and shortcomings. In this fast-moving scenario, the authors have worked rapidly to capture as best they could the current standard of information for users provided by sellers of home-sampling COVID-19 tests on their websites. Attention to details of dates accessed, etc., enables readers to know at which stage of the changing management and governance of the disease the information was available/accessible. The 13 basic, necessary information items chosen were each considered for all websites, with findings communicated in the text and in tables, enabling easy consideration by the reader. This format, and the authors` process for assembling their findings, were well described. Anomalies with respect to recommendations and advice about testing were brought to attention of readers. Essential basic information for readers, such as the difference between the two tests, and the different meanings of sensitivity and specificity versus positive and negative predictive value were set out; necessary, I believe, not just for citizens considering testing, but also perhaps for some general health practitioners who may not have had cause heretofore to be precise about using the correct terms. It was useful and interesting to be provided with verbatim quotations from some of the websites, clarifying for example, muddling of finger-prick as opposed to venous samples, and how they had become confused or obfuscated in the information for users. The `participants` in this study would be the potential purchasers of these home-sampling kits. Careful choice of the most appropriate
---

word to describe these people in a particular context is important, especially as the paper is exploring the quality of commercial websites selling a product. Avoidance of the word `consumer` is I believe, desirable, and `patient` can be incorrect. (See my `minor comments` below for instances and suggestions.) This is even more important in the context of attempting to regulate commercial influences in Medicine.

The ethical aspects of presenting unbalanced, incomplete, inaccurate and persuasive information for citizens are important: profit for the supplier rather than the good of the purchaser perhaps being the motivation for lack of clarity and other shortcomings.

The results were clearly set out under sub-headings, and were well-referenced – as up-to-date as they could be under the pressured circumstances of a fast-moving global pandemic! Emphasis was given to such matters as the failure for the implications of positive and negative test results to be covered by any UK regulation: an important matter not only for those using the tests, but for general interpretation and use of resulting data, especially in the media, which then compounds the problem.

It was unfortunate but understandable that the very rapid timeframes did not allow for more public involvement in exploring this topic, especially as they are the target for sellers` information. It would be useful to undertake more in-depth exploration by means of well-facilitated qualitative research embedded in the quantitative. This type of collaboration was undertaken (in a very different setting) to great benefit in the Protect Trial. [3]

Some specific comments on the text:

Page 2. Conclusion of abstract: Last sentence: May I suggest `More independence from unregulated commercial influence should be encouraged and enabled through regulation of suppliers, to ensure provision of good quality information for users` or similar.

Page 8, line 45: `indicating time periods which are known to be...`
And: `Some websites stated dates based on time since exposure` perhaps needs clarification. Does it mean `...asked for dates...`?

Page 8, line 53. End of sentence `who are not readily accessible` could perhaps be better expressed, perhaps `impractical for many/most people` or similar.

Page 9. Line 18. End of paragraph: `... noting how applicable evidence is to the public`. Needs clarification.

Line 42. `To interpret results, test users...` Perhaps not just `test users`? But others too, including involved health professionals as well?

Page 9. Last heading, No. 5. Suggest perhaps something along the lines of: `Consequences and implications for decision-making resulting from poor quality information for users of tests`.

Page 10, Line 4. Perhaps: `trustworthy` guidance?

Page 10, last sentence: Suggest: `This should be underpinned by robust collaborative qualitative research exploring how members of

the public interpret information and measures of accuracy, thus informing how it can be provided in a way that is clear, complete and accessible`.

Minor comments/typos:

Page 3. Line 53: `We chose`.

Page 7. Line 49: `...be different from those summarized here`.

Page 8. Line 20. Suggest perhaps insertion of a heading here: `Key communication requirements`.

Page 8. Line 21. Suggest `all test users` rather than `test users and patients`.

Page 9. Line 14. Rather than `not on patients`, perhaps `in a real-world`?

Page 9. Line 25 and 26. Suggest comma after UK websites and full stop after finger-prick samples. Then: `However, 2/18 remain...`. The following sentence needs clarifying, especially the end: `...the name and manufacturer could be identified to check for two tests`. Which two tests? Sorry, I found this confusing.

Page 16. Line 26. Substitute `potential purchaser` for `consumer`

Page 16. Line 28. Item 6. As it stands, omit `consumer`. But what is the assumption here based on, about preferences?

Page 16. Line 47. Omit `consumers` and substitute: `...and allow those interested to find out more`.

Page 16 and 17. Items 9, 10, 11, 12 and 13. Substitute `do they` with `does it`.

Page 22. Line 13. Typo: `Healthcare`.

Page 26. Omit `to the consumer` in description of graphic.

Reference was made to use of frequencies, rather than probabilities. Frequencies should always be used.

[1] Moynihan R, Macdonald H, Bero L, Godlee F. Commercial influence and covid-19. *BMJ*2020;369:m2456

[2] Fiona Godlee Editor`s choice 26th June: Covid 19: Where`s the strategy for testing?
BMJ 2020; 369 doi: <https://doi.org/10.1136/bmj.m2518> (Published 26 June 2020) *BMJ* 2020;369:m2518

[3] Donovan J, Mills N, Brindle L, Frankel S, Smith M, Jacoby A, et al. Improving the design and conduct of randomised trials by embedding them in qualitative research: the ProtecT study. *BMJ* 2002;325:766-70

REVIEWER	Stuart Hogarth Lectuer University of Cambridge
REVIEW RETURNED	29-Jun-2020

GENERAL COMMENTS	This article is timely, addresses a topic of major public health importance, and should be published as soon as possible. Although using a well-established method for the study of consumer testing services, it is highly original in its focus on the new market for COVID-19 tests. The research method and results are clearly described, and the analysis provides a thorough discussion of the areas of greatest concern. Given that the researchers have used a comparative two-country approach, it would have been interesting to see some more reflection on the contrasting situations in these two countries. I realise that price was not a key issue for the study but some further discussion of the significantly higher costs in the UK would have been welcome. The discussion of regulatory controls could be enhanced. At the outset, the authors point out the regulatory requirements that are specific to tests for lay users. This category of IVD is one of the few that under the current EU regulations has to go through a Notified Body. Regulation of product labelling provides a means to stipulate the categories of data that must be provided to consumers. There is currently no equivalent for online testing services. This article clearly illustrates the need to address the inconsistency between self-testing products and self-testing services and this point might be brought out in the discussion.
---

VERSION 1 – AUTHOR RESPONSE

Reviewer comments	Response and Revisions
Reviewer: 1 Margaret McCartney Comments: page 2 - instead of saying 'websites' should indicate type of test/limitations, would be better to say 'companies' so that better info is recommended to be included in all sales points	Thank you, we have changed this in the strengths and limitations section on page 2 and in the discussion on page 8.
page 3 ref 2 - This says higher sensitivities https://www.bmj.com/content/369/bmj.m2066 for antibody testing - if disagree helpful to say why	Ref 2 is for the PCR tests. Ref 3 is for the antibody tests and we say 80-90% sensitivity. The reference you cite gives 87% and 94% at the time point when the test is most accurate, so we do broadly agree with this reference.
page 4 - It would be really helpful to know the proportion of online sales which did make people go through some kind of assessment with video/healthcare professional first. I understand why not included but v useful information to know more broadly ie are these tests in general being sold with no extra information (a follow up would be really useful to find how much better these sets	We agree this is interesting, thank you for highlighting. We have added the following text at the beginning of the results section ““For the UK our Google searches retrieved 550 results, and for the US they retrieved 430 results. After the first round of sifting by 2 reviewers 46 potentially eligible websites were

of information are...)	identified. Of these 19 websites were later excluded, 13 of which only sold in quantities greater than one or to laboratories/hospitals/workplaces, 5 who incorporated contact with a health professional before the sale, and one which was withdrawn from sale between the search and extraction.” So in answer to your question we identified 5 with contact before purchase, and 41 with no contact. This ratio is probably reasonably representative, because we didn’t exclude on the basis of contact with a healthcare professional at the search stage.
Box 1 - am not sure that the example of better communication for 'who should take the test' really is much clearer. It needs framed in days surely - appreciate text abbreviated. problem is that gov information is currently worse ... ideally I think this should say eg 'Testing too early or too late can give a negative test even when you do have the virus. The best time to test is.... however... 'etc. may be better to say what points need stated in order to give fair info. Appreciate this is not going to appear very often on sales sites but by explaining what is ideal/needed I think will help ASA in particular when they review their guidance.	Thank you very good point, the example we gave was less misleading but not very informative. Changed as follows: ““Ideally samples should be taken from symptomatic individuals between days 1-5 from symptom onset. However, there are many cases when virus can be detected later into the illness.”²⁰ It would also be helpful to communicate that taking the test too early or late when it is less accurate may result in the test missing COVID-19 when it is present.”
second on 'test accuracy' - this relates to antibody testing from the reference and is not a test for current covid-19 as per the example of better communication. needs a top line eg "all tests have some degree of inaccuracy. If you have a negative result (you have not got antibodies to covid-19) then the test is very likely to be correct. If you get a positive result (you have got antibodies to covid-19) then the result is less accurate.' also worth saying that 'accuracy' is a statistical term but ? is it being used in the statistically correct way here ?	Thank you very much, we like this and have incorporated it as follows: No website provided a full explanation of accuracy, we suggest our own example as follows “Test accuracy: The tests are sometimes inaccurate. If you have a negative result (you have not got antibodies to covid-19) then the test is very likely to be correct. If you get a positive result (you have got antibodies to covid-19) then the result is less accurate. Of the people who test positive, 92 in 100 do actually have COVID-19. Of the people who test negative, more than 99 in 100 do not have COVID -19. Here is more detail on the science: Test accuracy was measured in an independent evaluation of 158 people with COVID-19 and 364 people without COVID-19].⁵⁴ The test had sensitivity 98% and specificity 99.2%. That means that if 1000 people are tested, and 100 of those have

	COVID-19, then 98 of the 100 people with COVID-19 will be detected and 2 will be missed (test negative). Of the 900 people who don't have COVID-19, 892 will test negative, and 8 will test positive (and believe they have COVID-19 when they do not).⁵⁴
in the other boxes there are still things I'd want to know as a customer (ie If the tests find covid-19, then it is very accurate. If the test is negative for covid-19, then it is very unreliable. Up to a third of people who do have covid-19 will test negative even though they do have the disease. and top line	We have added in the boxes on interpreting test results of molecular virus tests, under the quote which recommends test negatives with symptoms continue self isolation we have added “Linking information on the low negative predictive value of the PCR test to recommendations to continue self-isolation may strengthen the message.”
Antibody tests are not very helpful for most people	We agree that the intended use of antibody testing is a really important topic, which merits a whole paper itself. In this paper because we are focusing on communication we have limited ourselves to clear communication that we do not yet know how antibody presence relates to immunity.
The only other thing I'd suggest is a box explaining kite marks/ CE marks, what needs MHRA approval , and not - it's very confusing and a clear brief explanation would be useful - this can relate to who needs to do what in order to make it better (am always alarmed at burden regulation ASA - a voluntary org who have to buy in expertise - are relied on for sorting out medical claims)	Thank you for this suggestion, we agree its confusing. We haven't made a box, because we don't want to confuse further by expanding out the explanation beyond the specific circumstance of selling direct to consumer COVID-19 tests via websites. Instead we have strengthened our explanation in the introduction and added to the discussion as follows: Added to introduction: “Most websites selling COVID-19 tests would be classified by the MHRA as ‘distributors’, which gives clear obligations to supply the information provided by manufacturers with the test, but no specific guidance around communication on the website at the point of sale. Such claims are covered by the Advertising Standards Agency” Added to the discussion: “Regulation of product labelling provides a means to oversee information communicated for self-testing products bought in person, but there is currently no equivalent for online testing services in the UK. This gap in regulation

	could be solved by expanding the responsibility of the MHRA to include communication by 'distributors' at the point of online purchase, working collaboratively with the Advertising Standards Agency."
Reviewer: 2 Ian S Watt Comments: This was a well written manuscript describing a cross-sectional observational study which had assessed the accuracy and completeness of information provided by websites selling home self-sampling and testing kits for COVID-19. Whilst previous studies have considered the accuracy and utility of information on health related websites, to my knowledge this is the first to do so for COVID-19 testing kits. The study findings are of importance for research, health professional, policy and lay audiences.	Thank you
I would support publication of the paper and have only a few minor suggestions to make re clarifying some of the detail in the paper. These are outlined below: * in the first sentence of the final paragraph of the introduction I am unclear as to whether the "home sampling" refers only to molecular virus testing or to antibody testing as well.	Thank you, we have changed to "We analysed the information given to individuals considering purchasing a molecular virus or antibody COVID-19 test online for home self-sampling.
*I was unclear what a "representative sample of websites" meant in the description of methods	Representative refers to representative of what a member of the public may see if they performed a search that day. "The search was designed to identify a representative sample of websites and online advertisements which would be seen by an individual searching for a non-specific COVID-19 test. We aimed to identify websites selling home self-sampling and testing for COVID-19 using molecular virus and/or antibody tests directly to users."
* In the results it would be helpful to have more detail on how the number of results in the Google searches (550 in the case of the UK) relates to the number of websites and in turn the number of tests identified	We have added this as follows: "For the UK our Google searches retrieved 550 results, and for the US they retrieved 430 results. After the first round of sifting by 2 reviewers 46 potentially eligible websites were identified. Of these 19 websites were later excluded, 13 of which only sold in quantities greater than one or to laboratories/hospitals/workplaces, 5 who incorporated contact with a health

	professional before the sale, and one which was withdrawn from sale between the search and extraction. We identified 23 molecular virus testing services and 18 antibody testing services meeting the inclusion criteria, sold via 27 websites (25 from the UK and 2 from the US).”
* I am unclear as to if any of the tests had been purchased whether further information might have been forthcoming when the tests were dispatched	We agree we do not know this, we expect some information in the instructions for use and beyond. Our primary interest was information available before making the purchasing decision, as we consider it necessary to provide sufficient information to consider whether to purchase. In particular where receiving the correct information will result in the individual not making the purchase. We have added the following to the discussion “We only assessed information provided prior to purchase, as complete information should be given at this stage to inform the purchasing decision. However, further information would have been given after purchase, for example within the instructions for use, which was beyond the scope of this paper.”
* In the conclusion the authors mention a number of entirely reasonable recommendations. Do the authors have any suggestions who should be responsible for them - for example who should develop best practice guidance for communication about tests to the public.	Thank you, this gave us pause for thought. We would suggest the MHRA as they have the regulatory powers, and they will already work with the manufacturers on the packaging and product information, working with the Advertising standards agency who have relevant expertise. Added to the discussion “Regulation of product labelling provides a means to oversee information communicated for self-testing products bought in person, but there is currently no equivalent for online testing services in the UK. This gap in regulation could be solved by expanding the responsibility of the MHRA to include communication by ‘distributors’ at the point of online purchase, working collaboratively with the Advertising Standards Agency.”
Reviewer: 3 Hazel Thornton Comments: This analysis of accuracy and completeness of user information provided by sellers’ websites selling home self-sampling COVID-19 tests is an original and important piece of work that clearly illustrates the pitfalls of a poorly regulated marketplace. It illuminates the urgent need for	Thank you. With respect to regulatory reform we have added this to the discussion as follows “Regulation of product labelling provides a means to oversee information communicated for self-testing products bought in person, but there is currently no equivalent for online testing services in the UK. This gap in

regulatory reform, ultimately enabling provision of trustworthy guidance to users of websites seeking commercial home testing kits – another small step to achieving more independence from unregulated commercial influences. [1] [2] I suggest this should be emphasized. The manuscript is well organized and generally clearly written. [1] Moynihan R, Macdonald H, Bero L, Godlee F. Commercial influence and covid-19. BMJ2020;369:m2456 [2] Fiona Godlee Editor`s choice 26th June: Covid 19: Where`s the strategy for testing? BMJ 2020; 369 doi: https://doi.org/10.1136/bmj.m2518 (Published 26 June 2020) BMJ 2020;369:m2518	regulation could be solved by expanding the responsibility of the MHRA to include communication by ‘distributors’ at the point of online purchase, working collaboratively with the Advertising Standards Agency.”
This piece of work will add to the accumulating information being published in the BMJ, both concerning the Covid-19 pandemic and its campaign to tackle commercial influences in health provision. It will be of particular interest to citizens, health professionals and policymakers. It has been methodically researched by a strong team to a pre-defined plan formulated to capture all the information that a purchaser would require in order to determine a test`s benefits and shortcomings.	Thank you
In this fast-moving scenario, the authors have worked rapidly to capture as best they could the current standard of information for users provided by sellers of home-sampling COVID-19 tests on their websites. Attention to details of dates accessed, etc., enables readers to know at which stage of the changing management and governance of the disease the information was available/accessible.	Thank you
The 13 basic, necessary information items chosen were each considered for all websites, with findings communicated in the text and in tables, enabling easy consideration by the reader. This format, and the authors` process for assembling their findings, were well described. Anomalies with respect to recommendations and advice about testing were brought to attention of readers. Essential basic information for readers, such as the difference between the two tests, and the	Thank you

different meanings of sensitivity and specificity versus positive and negative predictive value were set out; necessary, I believe, not just for citizens considering testing, but also perhaps for some general health practitioners who may not have had cause heretofore to be precise about using the correct terms. It was useful and interesting to be provided with verbatim quotations from some of the websites, clarifying for example, muddling of finger-prick as opposed to venous samples, and how they had become confused or obfuscated in the information for users.	
The `participants` in this study would be the potential purchasers of these home-sampling kits. Careful choice of the most appropriate word to describe these people in a particular context is important, especially as the paper is exploring the quality of commercial websites selling a product. Avoidance of the word `consumer` is I believe, desirable, and `patient` can be incorrect. (See my `minor comments` below for instances and suggestions.) This is even more important in the context of attempting to regulate commercial influences in Medicine.	Yes we agree, we struggled with this. Thank you for providing suggestions, we respond to each instance below.
The ethical aspects of presenting unbalanced, incomplete, inaccurate and persuasive information for citizens are important: profit for the supplier rather than the good of the purchaser perhaps being the motivation for lack of clarity and other shortcomings. The results were clearly set out under sub-headings, and were well-referenced – as up-to-date as they could be under the pressured circumstances of a fast-moving global pandemic! Emphasis was given to such matters as the failure for the implications of positive and negative test results to be covered by any UK regulation: an	We agree with the need for qualitative research in this area. We have changed the last sentence of the discussion to reflect this as you suggest below. We recognise that the point about the ethical aspects is also important, and the conflict between profits and communication of correct information is the reason we have suggested that this area requires regulation.

important matter not only for those using the tests, but for general interpretation and use of resulting data, especially in the media, which then compounds the problem. It was unfortunate but understandable that the very rapid timeframes did not allow for more public involvement in exploring this topic, especially as they are the target for sellers' information. It would be useful to undertake more in-depth exploration by means of well-facilitated qualitative research embedded in the quantitative. This type of collaboration was undertaken (in a very different setting) to great benefit in the ProtecT Trial. [3] [3] Donovan J, Mills N, Brindle L, Frankel S, Smith M, Jacoby A, et al. Improving the design and conduct of randomised trials by embedding them in qualitative research: the ProtecT study. BMJ 2002;325:766-70	
Some specific comments on the text: Page 2. Conclusion of abstract: Last sentence: May I suggest 'More independence from unregulated commercial influence should be encouraged and enabled through regulation of suppliers, to ensure provision of good quality information for users' or similar.	Thank you. Although we do fully agree with this point, we have not made this change because we are trying to keep the style dispassionately scientific. Instead we have added to the discussion as described above.
Page 8, line 45: 'indicating time periods which are known to be...' And: 'Some websites stated dates based on time since exposure' perhaps needs clarification. Does it mean '...asked for dates...'?	Thank you we have changed to the following: "Recommended time points when samples should be taken were absent for 10/39 UK tests (26%). Some timing statements were misleading, suggesting using the test at time points which are known to be too early or too late. Some websites stated dates based on time since exposure, others since symptom onset which is median 5 days after exposure. Both are required to be able to advise both asymptomatic patients and patients with unknown exposure when they should order and use the tests."
Page 8, line 53. End of sentence 'who are not readily accessible' could perhaps be better expressed, perhaps 'impractical for many/most people' or similar.	Thank you – changed as suggested
Page 9. Line 18. End of paragraph: '... noting how applicable evidence is to the public'. Needs clarification.	This refers back to the earlier statement in the paragraph about analytic validity studies. I've added clarification earlier in the paragraph

	“Accuracy measures from analytic validity studies should not be assumed to give a good representation of test accuracy when applied in practice to the public.” Which links into the revised last sentence “It is important that the reported accuracy is based on all reviewed evidence and not selected results, and clearly explains how applicable the evidence is to the public.”
Line 42. `To interpret results, test users...` Perhaps not just `test users`? But others too, including involved health professionals as well?	We agree but we have kept the focus on test users as we are examining direct to consumer sales of tests, without involvement of healthcare professionals,
Page 9. Last heading, No. 5. Suggest perhaps something along the lines of: `Consequences and implications for decision-making resulting from poor quality information for users of tests` .?	We have aimed for five simple headings to describe the issues, and have kept the heading as is to match the format of the previous four. Your comment has highlighted the issue that we haven't provided sufficient explanation of this category. We have added “Clear communication about the meaning of test results as detailed above should be linked to evidence-based guidance about behaviour modification in light of test results. We found widespread evidence of websites failing to provide such evidence-based guidance, and some cases of websites actively suggesting unsafe behaviour.”
Page 10, Line 4. Perhaps: `trustworthy` guidance?	Changed as suggested
Page 10, last sentence: Suggest: `This should be underpinned by robust collaborative qualitative research exploring how members of the public interpret information and measures of accuracy, thus informing how it can be provided in a way that is clear, complete and accessible` .	Changed as suggested
Minor comments/typos: Page 3. Line 53: `We chose` .	Corrected, thank you.
Page 7. Line 49: `...be different from those summarized here` .	Changed as suggested
Page 8. Line 20. Suggest perhaps insertion of a heading here: `Key communication requirements` .	Changed as suggested
Page 8. Line 21. Suggest `all test users` rather than `test users and patients` .	Changed as suggested
Page 9. Line 14. Rather than `not on patients`, perhaps `in a real-world` ?	We explicitly mean patients here, their samples are made up of diluted spiked sera,

	whereas we want the denominator of accuracy studies to be people not derived samples.
Page 9. Line 25 and 26. Suggest comma after UK websites and full stop after finger-prick samples. Then: `However, 2/18 remain...`. The following sentence needs clarifying, especially the end: `...the name and manufacturer could be identified to check for two tests`. Which two tests? Sorry, I found this confusing.	Changed as suggested, and clarified as follows: "The molecular tests we could identify are approved for home sampling, however, the name and manufacturer was not identifiable for most websites."
Page 16. Line 26. Substitute `potential purchaser` for `consumer`	Done, and consumer removed/changed throughout the table.
Page 16. Line 28. Item 6. As it stands, omit `consumer`. But what is the assumption here based on, about preferences?	Changed as suggested
Page 16. Line 47. Omit `consumers` and substitute: `...and allow those interested to find out more`.	Changed as suggested
Page 16 and 17. Items 9, 10, 11, 12 and 13. Substitute `do they` with `does it`.	"Do they" replaced with "Does the website" throughout the table for clarity.
Page 22. Line 13. Typo: `Healthcare`.	Changed as suggested
Page 26. Omit `to the consumer` in description of graphic.	Changed as suggested
Reference was made to use of frequencies, rather than probabilities. Frequencies should always be used.	Thank you.
Reviewer: 4 Stuart Hogarth Comments: This article is timely, addresses a topic of major public health importance, and should be published as soon as possible. Although using a well-established method for the study of consumer testing services, it is highly original in its focus on the new market for COVID-19 tests.	Thank you for your kind comments.
The research method and results are clearly described, and the analysis provides a thorough discussion of the areas of greatest concern. Given that the researchers have used a comparative two-country approach, it would have	Thank you. We had intended to expand more on the comparison between countries in the discussion, but were reticent to say too much because we only found two tests from the US. That in itself does indicate are more stringent

been interesting to see some more reflection on the contrasting situations in these two countries. I realise that price was not a key issue for the study but some further discussion of the significantly higher costs in the UK would have been welcome.	regulatory approach in the US at the time of data extraction, although this has changed over time in both countries. We were also conscious of word count because we allowed space for consideration of what should be communicated in the discussion. We agree it is interesting that the UK prices tended to be higher than the US, but again we are reluctant to draw any firm conclusions on that due to the small sample in the US. We have added some consideration of price variation as follows: “There was a large variation in price of testing in the UK, and in many cases these differences do not appear to be justified by differences in the service provided. Greater regulation and standardisation of website claims may reduce this price differential by making comparisons between websites easier, and removing unsubstantiated claims.”
The discussion of regulatory controls could be enhanced. At the outset, the authors point out the regulatory requirements that are specific to tests for lay users. This category of IVD is one of the few that under the current EU regulations has to go through a Notified Body. Regulation of product labelling provides a means to stipulate the categories of data that must be provided to consumers. There is currently no equivalent for online testing services. This article clearly illustrates the need to address the inconsistency between self-testing products and self-testing services and this point might be brought out in the discussion.	Thank you this is a really insightful observation, we have added to the discussion “Regulation of product labelling provides a means to oversee information communicated for self-testing products bought in person, but there is currently no equivalent for online testing services in the UK. This gap in regulation could be solved by expanding the responsibility of the MHRA to include communication by ‘distributors’ at the point of online purchase, working collaboratively with the Advertising Standards Agency.”

VERSION 2 – REVIEW

REVIEWER	Margaret McCartney GP Glasgow
REVIEW RETURNED	18-Aug-2020

GENERAL COMMENTS	This is such a great paper and should have a big impact on policy. It will be very useful to help the ASA and to hold the MHRA to account. I appreciate the systematic approach which has also been flexible and very thorough. I note the authors have made some changes and it reads very well. My only criticism is that this has not yet been published. I think it needs to be expedited. Many thanks for writing it.
---

REVIEWER	Ian Watt Hull York Medical School / Dept of Health Sciences, University of York UK
REVIEW RETURNED	31-Aug-2020

GENERAL COMMENTS	I am happy that the authors have satisfactorily responded to previous comments and have nothing further to add.
---

REVIEWER	Hazel Thornton Department of Health Sciences, University of Leicester UK
REVIEW RETURNED	21-Aug-2020

GENERAL COMMENTS	This original, timely and much needed report has been enhanced by appropriate adoption of the variety of comments from reviewers. I believe these findings should be quickly and thoroughly publicised in order that the public be alerted to the dangers that can result from purchasing testing kits which are not fit for purpose and in some cases harmful. It is important that adequate regulation is achieved - without delay or prevarication.
--